# First Record of *Ophidonais serpentina* (Müller, 1773) (Oligochaeta: Naididae) in China: The Occurrence or Absence of Needles Are Intraspecific Differences

**Jiefeng Yu** [1,2], **Tingting Zhou** [1,2], **Hongzhu Wang** [1] **and Yongde Cui** [1,*]

1 State Key Laboratory of Freshwater Ecology and Biotechnology, Institute of Hydrobiology, Chinese Academy of Sciences, Wuhan 430072, China; yujiefeng@ihb.ac.cn (J.Y.); ttzhou319@163.com (T.Z.); wanghz@ihb.ac.cn (H.W.)
2 College of Advanced Agricultural Sciences, University of Chinese Academy of Sciences, Beijing 100049, China
* Correspondence: ydcui@ihb.ac.cn

**Abstract:** A naidid oligochaete, *Ophidonais serpentina* (Müller, 1773) is redescribed based on specimens from the Xinkai River in Zhejiang Province, China. *O. serpentina* is very common in Europe and America. This study is the first record of the species in China. By integrating the previously morphological descriptions related to *O. serpentina* in the world, it can be divided into three morphological groups: a group with dorsal chaetae starting from VI, a group without dorsal chaetae, and a group with an unstable starting position of the dorsal chaetae. By comparing the mitochondrial DNA (16S rDNA, COI), nuclear DNA (ITS2), and histones (H3) from the three groups, Bayesian inference and maximum likelihood phylogenetic analyses were performed based on the combined data set. Different analyses gave almost consistent phylogenetic trees. All of the genetic distances between the three groups were 0.00%. No genetic variation can be detected between the specimens regardless of the presence and starting position of dorsal chaetae. This result suggests that a single lineage of *O. serpentina* is widespread worldwide.

**Keywords:** new records; *Ophidonais serpentina*; mitochondrial genes; nuclear genes; histone genes; phylogeny; taxonomy





## 1. Introduction

*Ophidonais serpentina* (Müller, 1773) (Annelida: Clitellata: Naididae: Naidinae) [1] is the type species of the monotypic genus *Ophidonais* Gervais [2]. In phylogeny, *O. serpentina* seems to be the sister group to a clade consisting of *Stylaria*, *Ripistes*, and *Arcteonais* [3]. This species is a common species in Europe, America, and Africa. Timm suggested that this species may be widely distributed in the Holarctic realm [4]. In Asia, it has previously been reported in Iran, Japan, Korea and Siberia. This study is the first to find the species in China.

In terms of morphology, dorsal chaetae have been previously described as stout and straight, beginning from VI [4,5]. In North America, Kathman and Brinkhurst found that in some specimens, all of the dorsal chaetae had been shed [6]. Ohtaka also noted the absence of dorsal chaetae [7]. Our specimens, which were collected from Xinkai River, could be divided into two groups. Most of the individuals had no needles, while the rest had needles with an uncertain beginning segment. By integrating records from other regions of the world, there are three morphological groups in *O. serpentina*, namely the no needles group, the stable needles group, and the unstable needles group.

The aim of the present study was to compare the morphological characteristics of these new specimens with corresponding features of the previously examined specimens of *O. serpentina*. Through the inclusion of molecular data, we explored whether the differences in morphology were related to genetics or represent phenotypic plasticity.

## 2. Materials and Methods

### 2.1. Taxon Sampling and Collection of Specimens

The Xinkai River is a eutrophic urban river in Shaoxing City, Zhejiang Province, China (29°95′99.06″ N, 120°47′57.55″ E) and a coastal river near the East China Sea. Specimens of *O. serpentina* were collected from the surface of hydrophytes at the shore of Xinkai River. A total of 64 specimens were collected, as shown in Table 1. The conditions of the site were as follows: the sediment was composed of silt and blocks, the water temperature was 25 °C, the depth was 0.5 m, the pH was 7.29, the conductivity was 428.5 μS/cm, the total dissolved solids (TDS) was 290.23 mg/L, the oxidation–reduction potential (ORP) was 564.25 mV, and the dissolved oxygen (DO) was 5.97 mg/L.

**Table 1.** Comparison for two morphological groups of specimens.

|  | **No Needles Group** | **With Needles Group** |
| --- | --- | --- |
| Specimens | 53 | 11 |
| Body length | 4–30 mm | 4–30 mm |
| N segments | 27–84 | 38–87 |
| First segment of needles | none | variable, beginning at segment between XIV to XXXXV, lacking in most of the segments |
| Ventral chaetae of II segment | length 150–178 μm, width 5 μm | length 150–178 μm, width 5 μm |
| Ventral chaetae of the remaining segment | length 95–125 μm, width 5 μm | length 95–125 μm, width 5 μm |
| Papillae | dorsal side, scattered, the beginning is variable most appear at the posterior segment; some individuals have abundant papillae, begin at X, then each segment has one papilla | few, only appear at those segments have no needles |

The specimens were preserved in 80–90% alcohol. The morphological characteristics were observed with a light microscope (LM) and a scanning electron microscope (SEM). The specimens used for SEM were acidified and dried naturally, then pasted onto the SEM copper platform with carbon conductive adhesive. After that, the specimens were sputtered with gold and placed under the SEM (Hitachi SU-8010) for observation and photography. The permanently preserved specimens were stained with borax carmine, separated by hydrochloric acid alcohol, gradient dehydrated by alcohol (70–99%), hyalinized with xylene, and sealed by Canadian gum. Drawings and measurement were based on preserved specimens. The specimens were deposited in the Institute of Hydrobiology, Chinese Academy of Sciences. In this study, six specimens were selected for the extraction of genes. The gene data of other related species were downloaded from the GenBank database. *Rhyacodrilus coccineus* and *Rhyacodrilus falciformis* were selected as outgroups (Table 2). Drawings were made with Adobe Photoshop CC 2019.

**Table 2.** Specimens and sequence used in this study, GenBank accession number and collection localities.

| Species | Collection Site or Source; Collector | 16S | COI | ITS2 | H3 |
| --- | --- | --- | --- | --- | --- |
| **Ingroup** | | | | | |
| *Allonais gwaliorensis* | Moat at Angor Wat, Cambodia; A. Ohtaka | KY633311 [3] | KY633391 [3] | KY633363 [3] | - |
| *Allonais inaequalis* | Pacaya-Samiria Reserve, Peruvian Amazon, Peru; D. Shain | DQ459952 [8] | KY633390 [3] | - | - |

**Table 2.** *Cont.*

| Species | Collection Site or Source; Collector | 16S | COI | ITS2 | H3 |
|---|---|---|---|---|---|
| *Allonais paraguayensis* | Ward's Natural Science (sold as Stylaria); A. E. Bely | GQ355399 [9] | AF534828 [10] | - | - |
| *Allonais pectinata* | East Lake Scenic Area of Wuhan, China; W. jiang | MN914711 [11] | MN935212 [11] | - | - |
| *Branchiodrilus hortensis* | Lake Tehang, Central Kalimantan, Indonesia; A. Ohtaka | KY633312 [3] | KY633393 [3] | KY633378 [3] | - |
| *Branchiodrilus semperi* | Pond in Bogor Botanical Garden, Bogor, West Java, Indonesia; A. Ohtaka | KY633315 [3] | KY633396 [3] | KY633379 [3] | - |
| *Chaetogaster diaphanus* | Lake Lången, Vårgårda, Sweden; C. Erséus | DQ459956 [8] | JQ519897 [12] | KY633380 [3] | - |
| *Chaetogaster limnaei* | Sacramento, CA, USA; A. Bely/J. Sikes | GQ355405 [9] | KF952355 [13] | - | - |
| *Chaetogaster diastrophus* | Hällekis, Götene, Sweden; C. Erséus | JQ424952 [12] | LT904771 [14] | - | - |
| *Dero borellii* | Experimental biofilter, Manchester Metropolitan Univ.,UK; M. Dempsey | KY633324 [3] | KY633385 [3] | KY633364 [3] | - |
| *Dero digitata* | Lake Lången, Vårgårda, Sweden; C. Erséus | DQ459954 [8] | KY633397 [3] | KY633381 [3] | MH744978 [15] |
| *Dero furcata* | Ditch, Fengshan, Kaohsiung, Taiwan; C.-R. Li | KY633325 [3] | KY633388 [3] | - | MH744979 [15] |
| *Dero superterrenus* | Eastern Melrose, Alachua Co., Fl., USA; D. Strom & M. Wetzel | KY633326 [3] | KY633389 [3] | - | - |
| *Nais barbata* | Lake Låttern, Vingåker, Sweden; C. Erséus | JQ424993 [12] | JQ519861 [12] | - | - |
| *Nais christinae* | Upper Kenai River, Moose beach, AK, USA; L. Arsan & S. Atkinson | JQ424969 [12] | JQ519824 [12] | - | - |
| *Nais stolci* | Charlottenlund, Ystad, Sweden; C. Erséus | JQ425026 [12] | JQ519894 [12] | - | - |
| *Ophidonais serpentina* 1 | Xinkai River of Shaoxing, China; T. T. Zhou & J. F. Yu | **OM033727** | **OM033378** | **OM033228** | **SRR17607623** |
| *Ophidonais serpentina* 2 | Xinkai River of Shaoxing, China; T. T. Zhou & J. F. Yu | **OM033728** | **OM033379** | **OM033229** | **SRR17607622** |
| *Ophidonais serpentina* 3 | Xinkai River of Shaoxing, China; T. T. Zhou & J. F. Yu | **OM033729** | **OM033380** | **OM033230** | **SRR17607621** |
| *Ophidonais serpentina* 4 | Xinkai River of Shaoxing, China; T. T. Zhou & J. F. Yu | **OM033730** | **OM033381** | **OM033231** | **SRR17607620** |
| *Ophidonais serpentina* 5 | Xinkai River of Shaoxing, China; T. T. Zhou & J. F. Yu | **OM033731** | **OM033382** | **OM033232** | **SRR17607619** |
| *Ophidonais serpentina* 6 | Xinkai River of Shaoxing, China; T. T. Zhou & J. F. Yu | **OM033732** | **OM033383** | **OM033233** | **SRR17607618** |
| *Ophidonais serpentina* A | Kungsbackaån River, Kungsbacka, Sweden; S. Kvist &M. Lindström | KY633327 [3] | KY633398 [3] | LN810239 [16] | - |
| *Ophidonais serpentina* B | San Francisco Creek, San Mateo Co., California, USA; S. Fend | DQ459939 [8] | LT903820 [14] | KY633367 [3] | - |
| *Ophidonais serpentina* C | Wildcat Creek, Richmond, CA, USA; A. Bely/J. Sikes | GQ355411 [9] | KY633398 [3] | KY633366 [3] | - |
| *Piguetiella blanci* | Lake Jäsen, Orsa, Sweden; M. Lindström; C. Erséus | KY633320 [3] | KY633402 [3] | KY633370 [3] | - |
| *Paranais botniensis* | Viken, Höganäs, Sweden; C. Erséus | KY633316 [3] | KY633399 [3] | KY633368 [3] | - |
| *Paranais frici* | Rappahannock River (brackish), Middlesex Co., VA, USA; S. Kvist | KY633318 [3] | KY633415 [3] | KY633369 [3] | - |
| *Paranais litorails* | Rhode River, Edgewater, MD, USA; A. Bely/J. Sikes | KY633319 [3] | KY633401 [3] | - | - |
| *Slavina appendiculata* | Lången Lake, near Alingsås, Västergötland, Sweden; C. Erséus | AY885582 [17] | KY633405 [3] | KY633371 [3] | - |

**Table 2.** *Cont.*

| Species | Collection Site or Source; Collector | 16S | COI | ITS2 | H3 |
|---|---|---|---|---|---|
| *Stylaria fossularia* | Kampong Chhnang, Lake Tonle Sap, Cambodia; A. Ohtaka | KY633322 [3] | KY633408 [3] | KY633374 [3] | - |
| *Specaria josinae* | Lake Lången, Vårgårda, Sweden; C. Erséus | KY633321 [3] | KY633407 [3] | KY633372 [3] | - |
| *Stylaria lacustris* | Lake Lången, Vårgårda, Sweden; C. Erséus | DQ459947 [8] | KY633409 [3] | KY633375 [3] | - |
| *Uncinais uncinata* | Lången Lake, near Alingsås, Västergötland, Sweden; C.Erséus | DQ459942 [8] | KY633410 [3] | KY633376 [3] | - |
| *Vejodovskyella comata* | Lången Lake, near Alingsås, Västergötland, Sweden; C. Erséus | AY885584 [17] | KY633411 [3] | KY633377 [3] | - |
| *Pristina aequiseta* | Paint Branch, College Park, MD, USA; A. Bely/J. Sikes | GQ355415 [9] | GQ355374 [9] | - | - |
| *Pristina leidyi* | Carolina Biological Supply (sold as Stylaria). | GQ355416 [9] | AF534853 [10] | - | - |
| *Pristina longiseta* | Lizard Island (freshwater), Great Barrier Reef, Queensland, Australia; C. Erséus | GU901850 [18] | GU902108 [18] | - | - |
| *Pristina minutus* | Tjärnö, Strömstad, Sweden; C. Erséus | DQ459958 [8] | KJ753865 [19] | - | - |
| *Haemonais waldvogeli* | Naolihe wuxinghu River of Helongjiang, China; T. T. Zhou | **OM264280** | **MW888774** | **MW885234** | - |
| **Outgroup** | | | | | |
| *Rhyacodrilus coccineus* | Lake Lången, Vårgårda, Sweden; C. Erséus | DQ459931 [8] | GU902110 [18] | - | KF267971 [13] |
| *Rhyacodrilus falciformis* | Vitärtskällan Spring, Kappelshamn, Gotland, Sweden; C. Erséus | DQ459938 [8] | KF267935 [13] | - | KF267970 [13] |

## 2.2. DNA Extraction, Amplification, and Sequencing

DNA was extracted by using a TIANamp Genomic DNA Kit and following the manufacturer's manual. The conditions for gene amplification are specified in Table 2. The 1μL amplified product was extracted and detected by gel electrophoresis. If the target bands were found, the polymerase chain reaction (PCR) products were sent to Icongene Ltd. (Wuhan, China) for sanger sequencing. All of the primers used for sequencing are described in Table 3.

**Table 3.** Primers and programs used for amplification and sequencing of fragments of the mitochondrial 16S and COI and nuclear ITS2 and H3 markers.

| Gene | Primer | Sequence 5′-3′ | The Program of PCR | Reference |
|---|---|---|---|---|
| 16S | 16SAR-L<br>16SBRH | CGCCTGTTTATCAAAAACAT<br>CCGGTCTGAACTCAGATCACGT | 30 s at 98 °C; 10 s at 98 °C; 45 s at 60 °C; 35 cycles of 1 min at 72 °C; 2 min at 72 °C. | Palumbi et al., 1991 [20]<br>Palumbi et al., 1991 |
| COI | LCO1490<br>HCO2198<br>COI-E | GGTCAACAAATCATAAAGATATTGG<br>TAAACTTCAGGGTGACCAAAAAATCA<br>TATACTTCTGGGTGTCCGAAGAATCA | 30 s at 98 °C; 10 s at 98 °C; 45 s at 45 °C; 35 cycles of 45 s at 72 °C; 3 min at 72 °C. | Folmer et al., 1994 [21]<br>Folmer et al., 1994<br>Bely and Wray 2004 [10] |
| ITS2 | 606F<br>1082R | GTCGATGAAGAGCGCAGCCA<br>TTAGTTTCTTTTCCTCCGCTT | 30 s at 98 °C; 10 s at 98 °C; 45 s at 55 °C; 35 cycles of 45 s at 72 °C; 3 min at 72 °C. | Liu, Erséus 2017 [22]<br>Liu, Erséus 2017 |
| H3 | H3F<br>H3R | ATGGCTCGTACCAAGCAGACVGC<br>ATATCCTTRGGCATKATRGTGAC | 5 min at 95 °C; 30 s at 95 °C; 30 s at 50 °C; 35 cycles of 90 s at 72 °C; 8 min at 72 °C | Brown et al., 1999 [23]<br>Brown et al., 1999 |

## 2.3. Alignments and Phylogenetic Analysis

We used BioEdit to check the original sequence, and then used SeqMan (DNASTAR) to connect the sequence when the sequence showed as single peak. We aligned the complete sequences into NCBI and did BLAST online. The newly acquired and downloaded sequences from the GenBank database were stored in FASTA format for further analysis. Then, we chose PhyloSuite [24] to analyze the data. Four sequences in batches were aligned

by using the '-auto' strategy and normal alignment mode in MAFFT [25] and aggregated into a sequence matrix. The best model for the BIC criterion was selected by a model finder [26]. The optimal model was chosen by BIC: GTR + F + G4. Under the conditions of the GTR + G + F model (2 parallel runs, $1 \times 10^7$ generations), Bayes 3.2.6 [27] was used to infer the occurrence of the Bayesian inference system, in which 25% of the initial sample data were discarded as burn-in. Maximum likelihood phylogenies were inferred using IQ-TREE [28] under the GTR + G4 + F model for 1000 standard bootstraps. MEGA X was used to calculate the genetic distance between two pairs, and a p-distance model was selected to complete the calculation [29]. We used the Interactive Tree of Life (https://itol.embl.de accessed on 2 March 2022) tool [30] and Adobe Illustrator 2021 to manipulate and combine the phylogenetic trees.

## 3. Results

### 3.1. Taxonomy and Morphology

***Ophidonais serpentina*** **(Müller, 1773)**

See Speber (1948) for synonymies [5].

**Description of Chinese specimens:** Length 4–30 mm, segments 27–87. The living individual is pale and has pigment stripes on its anterior segments, wrapped by a thin crust of foreign particles (Figure 1D). Coelomocytes are abundantly present (Figure 1C). Obtuse prostomium, eyes present (Figure 1B). Stomach slowly widening in VIII–IX. Unable to swim.

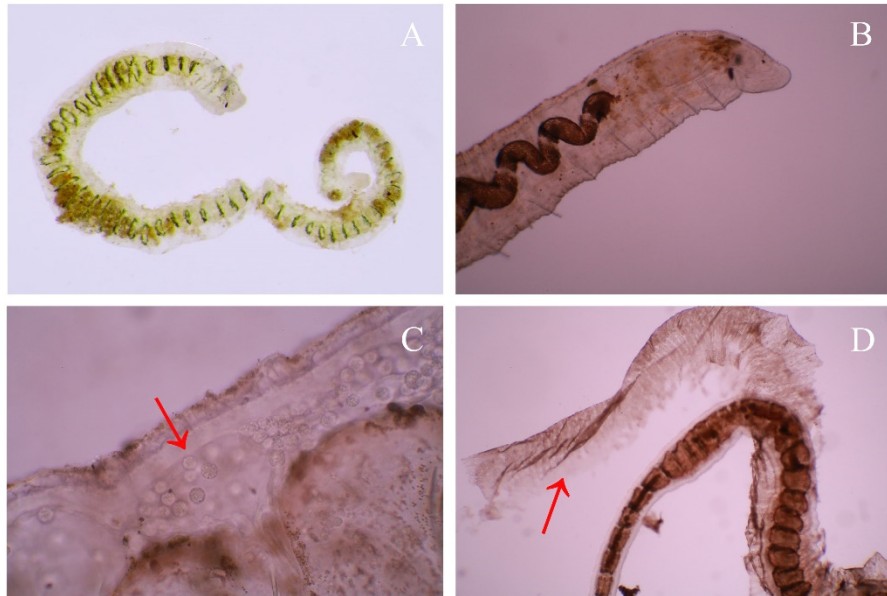

**Figure 1.** LM micrographs of *Ophidonais serpentina* specimens from China. (**A**) Complete individual, (**B**) eyes, (**C**) coelomocytes, and (**D**) thin crust of foreign particles. Observed from 4×, 20×, 40×, and 10×, respectively.

Needles are absent or beginning at posterior segments, often absent (Figures 2B and 3C–E). Needles are stout and straight, length 62–140 μm, width 5 μm, 1 per bundle, 2–3 equal pointed distal end, indistinct under the light microscope, proximal blunt, nodulus proximal 1/3–1/4, and indistinct. Ventral chaetae are 3–5 per bundle, with those of II longer than the rest, having a length 150–178 μm and width 5 μm; other ventral chaetae have length 95–125 μm and width 5 μm. The distal teeth of ventral chaetae are longer than the proximal teeth in anterior segments and equal in the following segments (Figures 2D–F and 3A,B). Nodulus median or proximal, nodulus distal gradually in the following segments.

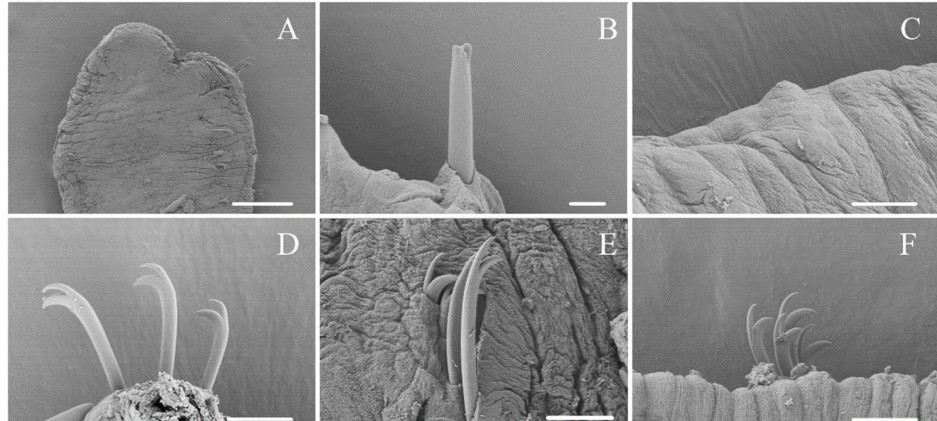

**Figure 2.** SEM micrographs of *Ophidonais serpentina*, specimens from China. (**A**) Peristomium; (**B**) needle; (**C**) papillae in the posterior of segments; and (**D**–**F**) ventral chaetae in segments VII, III, and V, respectively. Scale bars: (**A**) 100 μm, (**B**) 5 μm, and (**C**–**F**) 15 μm.

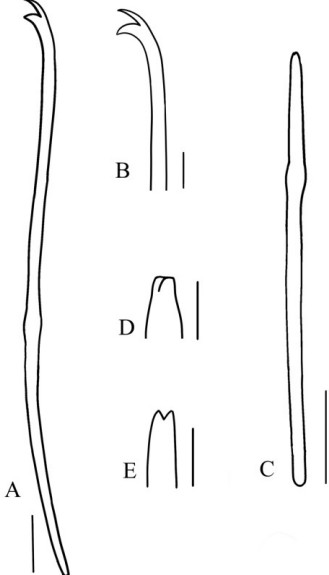

**Figure 3.** Chaetae of *Ophidonais serpentina*. (**A**) Anterior ventral, (**B**) posterior ventral, and (**C**) needles. (**D**,**E**) The end of needles. Scale bars: (**A**–**B**) 10 μm, (**C**) 20 μm, and (**D**,**E**) 5 μm.

**Specimens deposited.** IHB ZJ20210628a-b, two whole-mounted specimens, immature, no needles; IHB ZJ20210628c-d, two whole-mounted specimens, immature, with needles. Six specimens were used for extracting DNA and two specimens were used for electron microscopy. The rest of the specimens were dipped in 80–90% alcohol, and preserved in the Institute of Hydrobiology, Chinese Academy of Sciences.

*3.2. Phylogenetic Analyses*

We combined 119 nucleotide sequences into a dataset (3603bp) for phylogenetic analysis. The trees based on the combined dataset were largely consistent in the maximum likelihood (ML) and Bayesian inference (BI) analysis. Both trees were shown in an equidistant version (Figure 4), and the support values were given as BI posterior probabilities (pp) and ML bootstrap support (bs). In the selected specimens, *Ophidonais serpentina* 1–3 were the individuals without dorsal chaetae, and *O. serpentina* 4–6 were those with dorsal chaetae. *O. serpentina* A-C were downloaded from GenBank. Two phylogenetic trees highly supported that all of the *O. serpentina* specimens were recovered as a monophyletic clade (pp 1.00, bs 100). In maximum likelihood analysis, *O. serpentina* was recovered as sister to

the clade comprising four species: *Piguetiella blanci*, *Specaria josinae*, *Stylaria lacustris*, and *Stylaria fossularis* (bs 54). In Bayesian inference analysis, *O. serpentina* was sister to the clade comprising *Slavina appendiculata* and *Vejodovskyella comata* (pp 0.66). In both trees, all seven species formed a single clade (pp 0.99, bs 89).

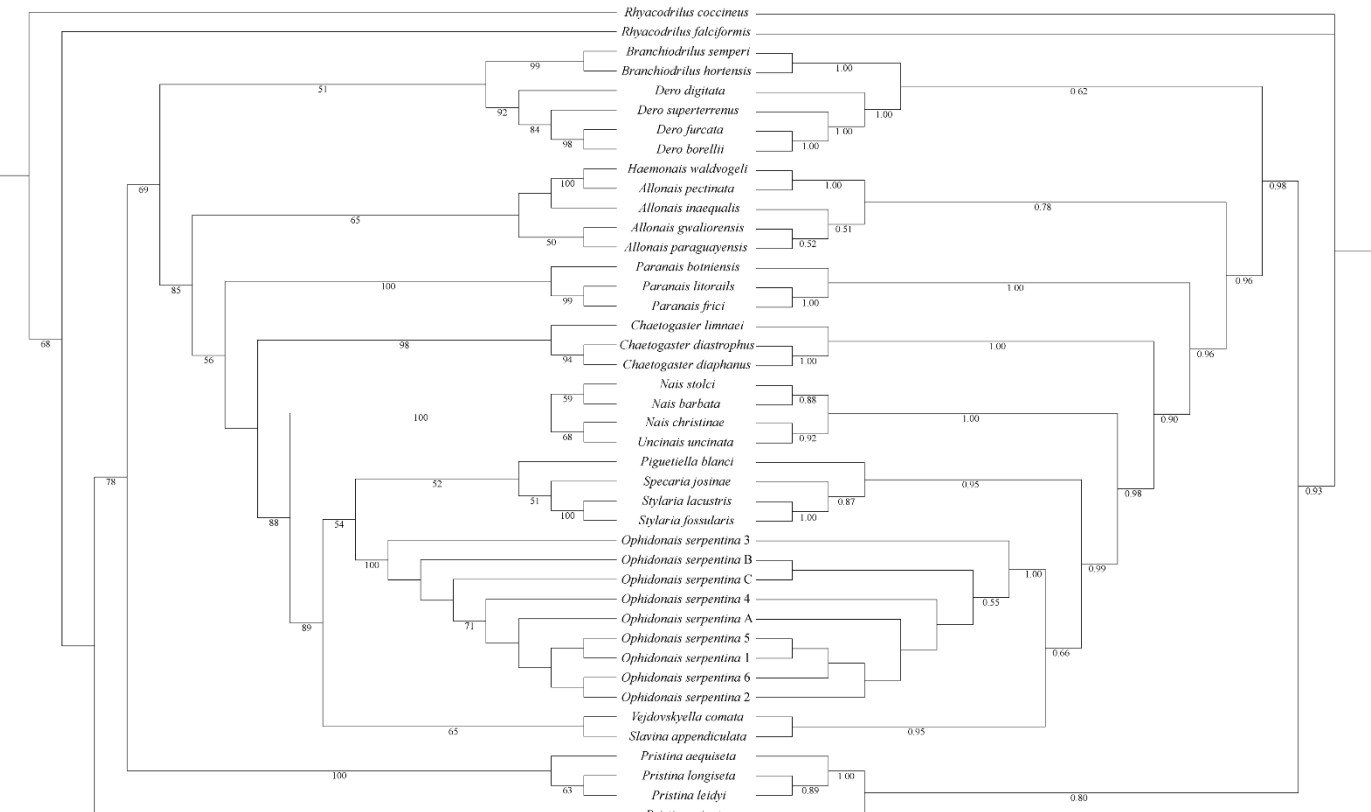

**Figure 4.** Phylogenetic trees obtained from the maximum likelihood (**left**) and Bayesian (**right**) analysis of the combined dataset.

### 3.3. Pairwise Genetic Distances

The uncorrected p-distance for 16S rDNA genes in the *Ophidonais* showed barcoding gaps were between 1.7 and 4.6%, and the COI genes were between 11.4 and 15.8%. The p-distances between the three morphological groups were 0.00% (Table 4).

**Table 4.** Genetic distances of *O. serpentina* with allied species (15 sequences) based on 16S gene (up) and COI gene (down).

| | 1 | 2 | 3 | 4 | 5 | 6 | 7 | 8 | 9 | 10 | 11 | 12 | 13 | 14 | 15 |
|---|---|---|---|---|---|---|---|---|---|---|---|---|---|---|---|
| 1. *Slavina appendiculata* | | 0.024 | 0.029 | 0.033 | 0.035 | 0.040 | 0.033 | 0.035 | 0.032 | 0.028 | 0.032 | 0.032 | 0.032 | 0.034 | 0.031 |
| 2. *Vejdovskyella comata* | 0.131 | | 0.021 | 0.031 | 0.031 | 0.033 | 0.026 | 0.028 | 0.026 | 0.025 | 0.026 | 0.026 | 0.030 | 0.027 | 0.029 |
| 3. *Piguetiella blanci* | 0.150 | 0.131 | | 0.019 | 0.023 | 0.023 | 0.029 | 0.031 | 0.029 | 0.023 | 0.029 | 0.029 | 0.027 | 0.029 | 0.029 |
| 4. *Specaria josinae* | 0.142 | 0.142 | 0.114 | | 0.031 | 0.029 | 0.042 | 0.044 | 0.042 | 0.038 | 0.042 | 0.042 | 0.041 | 0.042 | 0.042 |
| 5. *Stylaria lacustris* | 0.157 | 0.134 | 0.126 | 0.143 | | 0.017 | 0.031 | 0.033 | 0.031 | 0.030 | 0.031 | 0.031 | 0.032 | 0.031 | 0.033 |
| 6. *Stylaria fossularis* | 0.158 | 0.132 | 0.138 | 0.134 | 0.114 | | 0.044 | 0.044 | 0.044 | 0.040 | 0.044 | 0.044 | 0.046 | 0.044 | 0.046 |
| 7. *Ophidonais serpentina* 1 | 0.132 | 0.137 | 0.120 | 0.128 | 0.131 | 0.147 | | 0.000 | 0.000 | 0.000 | 0.000 | 0.000 | 0.000 | 0.000 | 0.000 |
| 8. *Ophidonais serpentina* 2 | 0.132 | 0.137 | 0.120 | 0.128 | 0.131 | 0.147 | 0.000 | | 0.000 | 0.000 | 0.000 | 0.000 | 0.000 | 0.000 | 0.000 |
| 9. *Ophidonais serpentina* 3 | 0.132 | 0.137 | 0.120 | 0.128 | 0.131 | 0.147 | 0.000 | 0.000 | | 0.000 | 0.000 | 0.000 | 0.000 | 0.000 | 0.000 |
| 10. *Ophidonais serpentina* 4 | 0.132 | 0.137 | 0.120 | 0.128 | 0.131 | 0.147 | 0.000 | 0.000 | 0.000 | | 0.000 | 0.000 | 0.000 | 0.000 | 0.000 |
| 11. *Ophidonais serpentina* 5 | 0.137 | 0.143 | 0.124 | 0.131 | 0.134 | 0.151 | 0.000 | 0.000 | 0.000 | 0.000 | | 0.000 | 0.000 | 0.000 | 0.000 |
| 12. *Ophidonais serpentina* 6 | 0.139 | 0.144 | 0.126 | 0.131 | 0.136 | 0.147 | 0.000 | 0.000 | 0.000 | 0.000 | 0.000 | | 0.000 | 0.000 | 0.000 |
| 13. *Ophidonais serpentina* A | 0.131 | 0.136 | 0.120 | 0.128 | 0.130 | 0.148 | 0.000 | 0.000 | 0.000 | 0.000 | 0.000 | 0.000 | | 0.000 | 0.000 |
| 14. *Ophidonais serpentina* B | 0.133 | 0.138 | 0.122 | 0.127 | 0.130 | 0.147 | 0.000 | 0.000 | 0.000 | 0.000 | 0.000 | 0.000 | 0.000 | | 0.000 |
| 15. *Ophidonais serpentina* C | 0.131 | 0.136 | 0.120 | 0.128 | 0.130 | 0.148 | 0.000 | 0.000 | 0.000 | 0.000 | 0.000 | 0.000 | 0.000 | 0.000 | |

## 4. Discussion

### 4.1. Morphological Characters of Ophidonais Serpentina

Of the 64 specimens, only 11 individuals had dorsal chaetae which appeared on one side of the body and with one on each segment. So far, the descriptions of this species in the published records were mostly consistent with Sperber (1948) and Timm (2009), without mentioning the lack of dorsal chaetae. This phenomenon has been shown in reports from North America [6] and Japan [31] and was also found in specimens collected in China. Additionally, some scholars found cilia in the papillae [32] but they were absent in our specimens and the arrangement of papillae was irregular. More detailed descriptions of the differences between the species from different regions of the world are listed in Table 5.

**Table 5.** The differences in morphology of *Ophidonais serpentina* from different continents.

| Information | Asia | Europe | America |
| --- | --- | --- | --- |
| Body length (mm) | 4–30 | 6–36 | 6 |
| Segment | 27–126 | 23–51 | 62 |
| Eyes | present | present | present or absent |
| Ventral chaetae in | | | |
| length (μm) | 150–208 | 152–179 | 130–174 |
| no./bundle | 2–5 | 2–6 | 2–4 |
| teeth | distal teeth shorter and thinner | distal teeth shorter and thinner | equal, or distal teeth slightly longer |
| Ventral chaetae after | | | |
| length (μm) | 152–168 | 128–158 | 100–130 |
| no./bundle | 2–4 | 2–6 | 3–4 |
| teeth | distal teeth shorter and thinner in anterior, equal in posterior | distal teeth shorter than proximal teeth | equal, or distal teeth slightly shorter |
| Needle | | | |
| length (μm) | 140–172 | 150–168 | 130 |
| staring segment | VI, or absent, not clear in the specimens in China | VI | VI |
| no./bundle | 1 | 1 | 1 |
| no. of teeth | 1–2, sometimes 3 | 1–2 | saw-toothed |
| Penial chaetae | two teeth closed, and the end swell and sag | the end swell and sag | immature |
| Habitat | shallow lake, the surface of hydrophytes | freshwater | common in rivers in North America, or live as parasites |
| Reference | Ohtaka and Iwakuma 1993; this study | Sperber 1948 | Spencer et al., 1993 [33]; Conn et al., 1994 [34] |

### 4.2. Intraspecies Analysis

The results of pairwise distance and phylogenetic analysis showed that *Ophidonais serpentina* was related closely to *Vejdovskyella comata* (pp 89, bs 0.66, mean distance 2.7% (16S), 13.8% (COI)), which was consistent with Bely and Wray's conclusion. Additionally, we found that *Piguetiella blanci* (pp 0.99, bs 54, mean distance 2.8% (16S), 12.1% (COI)) was also closely related to *O. serpentina*, which was not mentioned in Bely's study. All the results showed that the three groups (the absent dorsal chaetae group, the stable dorsal chaetae group and the unstable dorsal chaetae group) from a single monophyly. Whether the needles existed or not, and whether the starting position of the needles was stable or not, there was no genetic variation that was detected between the specimens. This

result suggests that a single lineage of *O. serpentina* is widespread worldwide. Whether the difference is caused by environmental factors or genetic factors needs further studies.

*4.3. Distribution and Habitat*

According to the published reports, *Ophidonais serpentina* prefer to inhabit on hydrophytes, however, the populations living in the Jajrood River (Iran) and the St. Lawrence River (Canada) show a parasitic habit. In some studies, scholars have reported *O. serpentina* as parasites in the mantle cavity of bivalves and crabs [34–36]. George et al. [37] suggested that water conductivity was one of the primary causes of these phenomena. A similar situation presented in Naidids *Dero*, where the first present segment of dorsal chaetae is related to life stage. In free-living individuals, the presence of dorsal chaetae begins at segment V or VI and in parasitic individuals, dorsal chaetae start from the IV segment [38]. Andrews et al. suggested that *Dero* parasitize in frogs' urinary tract and reproduce asexually. When the frogs void bladder urine, they will leave the host and enter back into the water for free-living [39]. However, the specimens in our study did not show parasitism. All of them were collected in the littoral zone of the Xinkai River, an urban river with good growth of *Hydrilla verticillata*, numerous residents and serious organic pollution. Perhaps the life stage leads to the variation of *O. serpentina*'s needles. Further studies are required to understand more about the causes of this intraspecific difference.

**Author Contributions:** Conceptualization, Y.C. and H.W.; methodology, Y.C., J.Y. and T.Z.; software, J.Y.; validation, Y.C., H.W., T.Z. and J.Y.; formal analysis, J.Y.; investigation, Y.C., T.Z. and J.Y.; resources, Y.C.; data curation, T.Z. and J.Y.; writing—original draft preparation, J.Y.; writing—review and editing, H.W., Y.C. and T.Z.; visualization, J.Y.; supervision, Y.C. and H.W.; project administration, Y.C.; funding acquisition, Y.C. All authors have read and agreed to the published version of the manuscript.

**Funding:** This research was funded by the Second Tibetan Plateau Scientific Expedition and Research (STEP) program (Grant no. 2019 QZKK0304) and the National Natural Science Foundation of China (52039006).

**Institutional Review Board Statement:** All applicable international, national, and/or institutional guidelines for the care and use of animals were followed. The cited data have been published in NCBI.

**Informed Consent Statement:** Not applicable.

**Data Availability Statement:** The data presented in this study are available upon request from the corresponding author.

**Acknowledgments:** We highly acknowledge Wei Jiang for revising the manuscript, and for his useful advice on writing and phylogenetic analyses.

**Conflicts of Interest:** The authors declare no conflict of interest. The funders did not take part in the design, collection, analyses, or the writing of the manuscript.

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
