# Peer review of "First Record of Ophidonais serpentina (Müller, 1773) (Oligochaeta: Naididae) in China: The Occurrence or Absence of Needles Are Intraspecific Differences"

_diversity, doi:10.3390/d14040265_

Round 1

Reviewer 1 Report

This is the first record of the species in China, with showing DNA data, and it is worth publishing. On the other hand, it seems that inappropriate hypothesis setting lead incorrect conclusion.

Major point
Grouping of the specimens into three (no dorsal setae group, stable dorsal setae group and unstable 193 dorsal setae group) is not appropriate, because they are actually.Every individual in any group was collected from a single asexually population in China. All of them could be clones, and the DNA results support this point. In other words, distribution of dorsal chaetae can vary within single lineage. It is probable that the Chinese population has been recently introduced and an invasive lineage has been widespread worldwide. Your results do not automatically mean there are no other lineages in the species.

Minor points
L51 (TDS) was 290.23 g/L. It seems be too high. mg/L?  Were the environmental parameters measured for surface water or bottom water?

115-122 The synonym list looks unnecessary, or put only representative literature. For example "See Sperber (1948) for synonymies"

L128. Figure 1. It is unclear what each picture shows. Specimen in Figure 1A appears to have no tail. Square eye cannot be recognized from the Figure 1B. Put arrow in Figures C and D.

L136  Text said that "The distal teeth of ventral setae longer than the proximal teeth in anterior segments, equal in the following segments". The explanation does not match the Fig. 2D, E, F, all of which have seems to have shorter distal teeth. What is the differences among D, E, F in Figure 2. in different segment? or variations?

L147 Specimens examined  ? Specimens deposited   

L179 [18].  [5] is earlier than [18]

L179 lack of 179 dorsal setae is much more common in China
It seems be not correct because it is derived from  single collection from one population in China and variability is unknown. It is also true for European or American populations. Kathman and Brinkhurst (1999) said that "We have seen some specimens in which all the dorsal chaetae have been shed" 
Kathman DD, Brinkhurst RO (1999) Guide to the Freshwater Oligochaetes of North America. Revised version. Aquatic Resources Center, TN, USA, 264pp.

L186 L2 from the bottom of the Table. Ohtaka 1993 should be replaced Ohtaka and Iwakuma 1993

L201 I could not access the literatures 22 nor 23, but they are really parasitic (metabolically dependent upon the host)?   not merely symbiotic? 

Author Response

Dear reviewer,

Thank you for your suggestions, and your suggestions have enabled us to make further improvements on our paper. Our responses are given as follows.

Point 1: Grouping of the specimens into three (no dorsal setae group, stable dorsal setae group and unstable 193 dorsal setae group) is not appropriate, because they are actually.Every individual in any group was collected from a single asexually population in China. All of them could be clones, and the DNA results support this point. In other words, distribution of dorsal chaetae can vary within single lineage. It is probable that the Chinese population has been recently introduced and an invasive lineage has been widespread worldwide. Your results do not automatically mean there are no other lineages in the species.

Response: Your point does make sense. However, in morphological observation, there are three groups in Ophidonais sernpentina. These differences have no correlation with phylogenetic analysis. And we suggested there is no cryptic species in the species, because the sequences downloaded from NCBI can also form a single clade with our sequences. Perhaps the individuals we collected were clones, however, they did have differences in morphology.

Point 2: L51 (TDS) was 290.23 g/L. It seems be too high. mg/L?  Were the environmental parameters measured for surface water or bottom water?

Response: L53 It’s mg/L. We measured bottom water in our study.

Point 3: 115-122 The synonym list looks unnecessary, or put only representative literature. For example "See Sperber (1948) for synonymies"

Response: L117 We’ve revised it as suggested.

Point 4: L128. Figure 1. It is unclear what each picture shows. Specimen in Figure 1A appears to have no tail. Square eye cannot be recognized from the Figure 1B. Put arrow in Figures C and D.

Response: L125 I changed Figure 1A to a complete individual’s photo. As for Figure 2, in my opinion, the eyes seemed like a pair of rectangles. However, the description is too subjectivity, so I modified it (deleted “square” or “rectangular”). I add arrows in C and D.

Point 5: L136 Text said that "The distal teeth of ventral setae longer than the proximal teeth in anterior segments, equal in the following segments". The explanation does not match the Fig. 2D, E, F, all of which have seems to have shorter distal teeth. What is the differences among D, E, F in Figure 2. in different segment? or variations?

Response: L137 Perhaps the viewing angle leads to a misunderstanding. I changed the Fig. 2E for showing the setae better. D, E and F are the photos of ventral setae in â…¦, â…¢, â…¤ segment respectively.

Point 6: L147 Specimens examined? Specimens deposited   

Response: L142 We’ve revised it as suggested.

Point 7: L179 [18].  [5] is earlier than [18]

Response: L174 Ohtaka’s research in 1999 mentioned the absence of needles, but not in 1993. My expression was amphibolous, thus I deleted “first”.

Point 8: L179 lack of 179 dorsal setae is much more common in China
It seems be not correct because it is derived from single collection from one population in China and variability is unknown. It is also true for European or American populations. Kathman and Brinkhurst (1999) said that "We have seen some specimens in which all the dorsal chaetae have been shed" 

Response: The conclusion was not very correct. I modified the associative description. And I downloaded the book, however, I did not find the description that "We have seen some specimens in which all the dorsal chaetae have been shed".

Point 9: L186 L2 from the bottom of the Table. Ohtaka 1993 should be replaced Ohtaka and Iwakuma 1993

Response: L189 We’ve revised it as suggested.

Point 10: L201 I could not access the literatures 22 nor 23, but they are really parasitic (metabolically dependent upon the host)?   not merely symbiotic? 

Response: L206 The conclusions in these two articles said that Ophidonais serpentina showed parasitic behavior in some situations.

Sincerely,

Yongde Cui

State Key Laboratory of Freshwater Ecology and Biotechnology

Institute of Hydrobiology

Chinese Academy of Sciences

Wuhan 430072

Reviewer 2 Report

Dear authors, I liked to read your ms. The study is well-done and resolves the issue of phenotypic plasticity in this peculiar naidid. The presence/absence of dorsal setae within the same species does not seem to be indicative for taxonomic differention, as previously thought in several morphology-based studies. It might be related to a free-living or parasitic life stage, as evidence before in the naidid genus (Allo)Dero. I recommend to discuss this issue in further detail.

I made some corrections/recommendation in the attached pdf-file, as a guidance to the ms revision.

Author Response

Dear reviewer,

Thank you for embellishing my manuscript, and we accepted all the retouches and revised the unclear description (in 3.3 pairwise genetic distance). After reading the article A new African species of parasitic Dero (Annelida, Clitellata, Naididae) in the urinary tract of reed frogs, we felt enlightened and improved the discussion. Thank you for the suggestion, and your suggestions have enabled us to make further improvements on our paper. Our responses are given as follows.

Point 1: Similar situation as in the Naidids Dero, in which the presence of dorsal chaetae depends on life stage free-living or parasitic.

Response: Perhaps O. serpentina has the same habit as Dero rwandae, thus some populations were detected in the host. And the variation of habit could cause the variation of needles. However, our study has no evidence to prove this idea, it needs further studies to know more about the causes of this intraspecific difference.

Point 2: Meaning of “foreign”?

Response: L125 the “foreign” means the thin crust was made of external materials.

Sincerely,

Yongde Cui

State Key Laboratory of Freshwater Ecology and Biotechnology

Institute of Hydrobiology

Chinese Academy of Sciences

Wuhan 430072

Reviewer 3 Report

I have reviewed the manuscript “First record of Ophidonais serpentina (Müller, 1773) (Oligochaeta: Naididae) in China, occurrence or absence of needles belongs to intraspecific differences” in which the authors present the first records of O. serpentina from China, and test if morphological variation is linked to different species. The manuscript is well-written and the methods are good to fulfil the aims of the study. Both the introduction and discussion are relatively short and to the point. The methods are described in sufficient detail. The results are presented clearly, and all conclusions seems to be supported by the data, the figures are also clear. I short, I think it is a good study and I do not have much to add, I only have some minor comments, se below.

Introduction: please add what family (and perhaps subfamily) the species belongs to in the first paragraph. Perhaps also add a sentence ore two about its phylogenetic position, this is not necessary, but I think it would be nice with a sentence like: “O. serpentina seems to be the sister-group to a clade consisting of Stylaria, Ripistes and Arcteonais (Erséus et al. 2017 doi: http://dx.doi.org/10.1016/j.ympev.2017.07.016)” or similar.

Table 3: suggest change “exist or not” to “present or absent”

Figure 4: I would have liked to see the trees scaled with genetic distance if it is possible. Otherwise, I assume those trees are available in fig S4.

Appendix 1: would it be possible to add a reference for the study the sequences are from?

Appendix 2: I would like a remainder in the table legend for which O. serpentina have or lack needles. This would be more important if there were any genetic variation.

Author Response

Dear reviewer,

Thank you for your suggestions. Your suggestions have enabled us to make further improvements on our paper. Our responses are given as follows.

Point 1: Please add what family (and perhaps subfamily) the species belongs to in the first paragraph. Perhaps also add a sentence one or two about its phylogenetic position.

Response: We completed the taxonomy information of O. serpentina in the introduction. After reading the article you mentioned, I added the sentences about its phylogenetic position.

Point 2: Table 3: suggest change “exist or not” to “present or absent”

Response: We’ve revised it as suggested.

Point 3: Figure 4: I would have liked to see the trees scaled with genetic distance if it is possible. Otherwise, I assume those trees are available in fig S4.

Response: At the bottom of this email are the phylogenetic trees with genetic distance.

Point 4: Appendix 1: would it be possible to add a reference for the study the sequences are from?

Response: We added the references in Appendix 1.

Point 5: Appendix 2: I would like a remainder in the table legend for which O. serpentina have or lack needles. This would be more important if there were any genetic variations.

Response: We made corresponding marks in Appendix 2 to distinguish O. serpentina have or lack needles in each sequence.

Sincerely,

Yongde Cui

State Key Laboratory of Freshwater Ecology and Biotechnology

Institute of Hydrobiology

Chinese Academy of Sciences

Wuhan 430072

Reviewer 4 Report

Notes to the Author. 

The work is made on high technical level, and the diligent re-description of the taxon is valuable. However, the  different presence and distribution of dorsal chaetae can change in aging  an individual, and the problem could be resolved in much cheeper way by observation on a number of separate worms during their lifetime (a year?) in aquaria. 

Line 30. The species is reported also from many places of Siberia, and once from Korea (all this is Asia!)

Line 132. Dorsal setae (but why not chaetae?) and needles are the same thing. 

Line 180. Plural of the Latin word "cilium" is "cilia".

Line 193. All three groups were monophyletic - maybe you thought that the group of these three groups form a single monophylon?

Lines 199-200. Not hygrophytes but hydrophytes (=water plants).

Lines 200-201. Idea on parasitism in this species is wrong. In the case of Iran, Ardalan et al. (2011) figured a first end with numerous chaetae in both dorsal and ventral bundles beginning in II. In the case of America, George is referring to Conn (1994) (listed under No 21 but not referred to in your text), a conference thesis not seen by me but probably presenting either a similar misidentification or an accidental finding.  

Line 282. Why capital letters?

Appendix 1. Correct the geographical names "Bond in Bogor" and "San Fransisquito".

Author Response

Dear reviewer,

Thank you for your suggestions. Your suggestions have enabled us to make further improvements on our paper. Our responses are given as follows.

Point 1: Line 30. The species is reported also from many places of Siberia, and once from Korea (all this is Asia!)

Response: L32 We’ve revised it as suggested.

Point 2: Line 132. Dorsal setae (but why not chaetae?) and needles are the same thing.

Response: L137 After checking Zoology, we found using chaetae may be more accurate than setae. Thus, we replaced all the setae into chaetae. And needle is a kind of dorsal chaetae.

Point 3: Line 180. Plural of the Latin word "cilium" is "cilia".

Response: L175 We’ve revised it as suggested.

Point 4: Line 193. All three groups were monophyletic - maybe you thought that the group of these three groups form single monophyly?

Response: L199 We’ve revised it as suggested.

Point 5: Lines 199-200. Not hygrophytes but hydrophytes (=water plants).

Response: L204-205 We’ve revised it as suggested.

Point 6: Lines 200-201. Idea on parasitism in this species is wrong. In the case of Iran, Ardalan et al. (2011) figured a first end with numerous chaetae in both dorsal and ventral bundles beginning in II. In the case of America, George is referring to Conn (1994) (listed under No 21 but not referred to in your text), a conference thesis not seen by me but probably presenting either a similar misidentification or an accidental finding. 

Response: L206 We added the possibility that the parasitism may be an accidental finding in the discussion. As for the relationship between dorsal setae and life stage, one possibility has been pointed out. There are similar situations present in Dero. Dero rwandae was detected in frogs. They would like to leave the host after a period of asexually reproduce. Perhaps O. serpentina has the same habit, thus some populations were detected in host. And the variation of habit could cause the variation of needles. However, our research has no evidence to prove this idea, it needs further studies to know more about the causes of this intraspecific difference.

Point 7: Line 282. Why capital letters?

Response: L301-302 We’ve revised it into lowercase letters.

Point 8: Appendix 1. Correct the geographical names "Bond in Bogor" and "San Fransisquito".

Response: We’ve revised it as suggested.

Sincerely,

Yongde Cui

State Key Laboratory of Freshwater Ecology and Biotechnology

Institute of Hydrobiology

Chinese Academy of Sciences

Wuhan 430072

Round 2

Reviewer 1 Report

Your results do not automatically mean no other lineages exist in and near the species because not all populations you analyzed. It seems more accurate that "No genetic variation can be detected among material examined regardless of presence and starting position of needle chaetae. The result suggests that single lineage of Ophidonais serpentine has been widespread worldwide."

As for the book of Kathman and Brinkhurst, you may see the first version in 1998. Kathman and Brinkhurst (1999) I suggested is the revised version. In this book, hey said "We have seen some specimens in which all the dorsal chaetae have been shed" in p. 50.

Kathman DD, Brinkhurst RO (1999) Guide to the Freshwater Oligochaetes of North America. Revised version. Aquatic Resources Center, TN, USA, 264pp.

Author Response

Dear reviewer,

Thank you for the time and effort that they have put into reviewing the manuscript. Your suggestion made us improve our manuscript better and our responses are given as follows:

Point 1: Your results do not automatically mean no other lineages exist in and near the species because not all populations you analyzed. It seems more accurate that "No genetic variation can be detected among material examined regardless of presence and starting position of needle chaetae. The result suggests that single lineage of Ophidonais serpentina has been widespread worldwide."

Response: L20-22, L169-172, We’ve revised it as suggested.

Point 2: As for the book of Kathman and Brinkhurst, you may see the first version in 1998. Kathman and Brinkhurst (1999) I suggested is the revised version. In this book, they said "We have seen some specimens in which all the dorsal chaetae have been shed" in p. 50.

Response: We found the sentence after getting the right version. And the inaccurate sentence in our manuscript has been deleted. And we replenished some descriptions in the Introduction (L34-35, L156).
